# Development of a Facial Expression Scale Using Farrowing as a Model of Pain in Sows

**DOI:** 10.3390/ani10112113

**Published:** 2020-11-14

**Authors:** Elena Navarro, Eva Mainau, Xavier Manteca

**Affiliations:** 1Department of Animal and Food Science, School of Veterinary Science, Universitat Autònoma de Barcelona, Bellaterra, 08193 Barcelona, Spain; xavier.manteca@uab.cat; 2AWEC Advisors SL, Ed. Eureka, Parc de Recerca UAB, Bellaterra, 08290 Barcelona, Spain; eva.mainau@uab.cat

**Keywords:** facial expression, grimace scale, pain, sow

## Abstract

**Simple Summary:**

Pain evaluation using non-invasive indicators can be difficult in production animals. Some years ago, a group of scientists realized that it was possible to evaluate pain by just considering animals’ facial expressions. All animals have their own facial expressions, and the animal’s expressiveness affects how difficult it is to find facial zones to evaluate pain. Until today, facial expressions have never been studied in sows. Our group decided to use farrowing (sow parturition) as a pain model to evaluate the sows’ facial expressions. Five different facial expression zones were found in our study: Tension above eyes, Snout angle, Neck tension, Temporal tension and ear position, and Cheek tension. The five zones were studied and evaluated by eight observers after a training session, obtaining good reliability results, especially on Tension above eyes, Snout angle, and Neck tension. These good results suggest that the sow facial expression scale could be a good non-invasive indicator to evaluate pain in sows.

**Abstract:**

Changes in facial expression have been shown to be a useful tool to assess pain severity in humans and animals, but facial scales have not yet been developed for all species. A facial expression scale in sows was developed using farrowing as a pain model. Five potential facial zones were identified: (i) Tension above eyes, (ii) Snout angle, (iii) Neck tension, (iv) Temporal tension and ear position (v), and Cheek tension. Facial zones were examined through 263 images of a total of 21 sows at farrowing, characterizing moments of non-pain (19 days post-farrowing; score 0), moderate pain (time interval between the delivery of two consecutive piglets; score 1) and severe pain (during active piglet delivery; score 2). Images were evaluated by a “Silver Standard” observer with experience in sows’ facial expressions, and by a group of eight animal welfare scientists, without experience in it, but who received a one-hour training session on how to assess pain in sows’ faces. Intra- and inter-observer reliability of the facial expression ranged from moderate to very good for all facial expression zones, with Tension above eyes, Snout angle, and Neck tension showing the highest reliability. In conclusion, monitoring facial expressions seems to be a useful tool to assess pain caused by farrowing.

## 1. Introduction

Pain is a sensory and emotional experience that has significant effects on animal welfare, leading to a negative production impact [1]. The measurement of pain in animals is very complex, and it continues to be a critical issue in veterinary care and biomedical research [2].

Almost two hundred years ago, Darwin had already described the non-human animals’ capacity for expressing emotions such as pain through facial expression [3]. In the last decade, facial expressions have been studied rigorously in several animal species including rats, mice, rabbits, ferrets, horses, sheep, piglets, and seals [4,5,6,7,8,9,10,11,12,13,14]. Facial expressions have been shown to be a good pain indicator in most species, allowing one not only to detect pain, but also to grade its severity after a noxious stimulus [4,5,6,7,8,9,10,12,13,14]. Grimace scales are based on the Facial Action Coding System (FACS), which were initially developed by Ekman and Friesen [15]. The development of facial expression scales requires the ability to identify the change of specific facial action units (FAUs) when animals are in pain. At least three FAUs have been described for each pain scale and all scales have demonstrated high inter-observer reliability among observers, proving their accuracy and feasibility [6]. Pain scores assigned by observers to painful conditions in farm animals are usually influenced by gender, age, and/or profession of the observers [16,17]. Di Giminiani et al. [8] stated that the level of pig knowledge of the observers was not correlated to the pain scores assigned in the facial scale in piglets. Despite this, to our knowledge, the effects of the observer’s gender and age on the inter-observer reliability test performed in facial expression scales have not been previously studied.

Many of the facial scales studied in different species showed that orbital tightening is one of the easiest regions for observers to recognize and use to evaluate pain [4,5,8,9,10,12], focusing on eye aperture.

Pigs are a prey species, so, although it has not been proven, it seems likely that they would tend to not express pain or weakness, making pain recognition and evaluation incredibly difficult [1]. For this reason, limited pain indicators have been documented to evaluate whether an animal is suffering from pain in a specific moment [18]. The parturition in sows is associated with increased plasma cortisol concentrations, C-reactive protein, and Haptoglobin, which could be indicative of pain [19]. So far, effective and non-invasive pain indicators in sows during farrowing are based on behavioral and corporal indicators such as legging, pawing, arching, trembling, or tail tickling [19,20].

As mentioned before, piglet facial scales have been studied, and they seem to be a very good tool to assess pain, but until now, no facial scales have been developed in sows.

Farm animals can suffer from pain throughout their lives. All mammalian female farm animals give birth at least once in their lives, and it is often without the provision of either anesthesia or analgesia during the process [21]. As happens in other species, it has been demonstrated that farrowing itself is a painful moment for the sows [19,21,22], and a difficult farrowing may affect sow behavior through exhaustion, sickness, or pain [23], which can develop into deficient maternal behavior.

The aims of this study were to (i) develop a facial expression scale based on FAUs using farrowing as a model of pain, thus avoiding producing unnecessary pain, and (ii) evaluate the effect of the observers’ sex and knowledge of the pig production sector on their rating behavior. The grading scale for pain provided herein might represent a substantial advance in pain recognition and management in sows and may become a fast, easy-to-learn, and highly reproducible tool for farrowing sows.

## 2. Materials and Methods

### 2.1. Animals, Housing, and General Management

The experimental procedure was carried out on a commercial farm (Casa-Ramona; Sora, Barcelona, Spain) from April to July 2018. A total of twenty-one Danbred sows, seven gilts and fourteen small-sized multiparous sows (from 2nd to 4th parity), were randomly selected on the day of parturition. A total of six different weekly study replicates with one to five sows in each replicate were studied. Sows that farrowed at night, sows that showed signs of lesions, illness, or lameness, and sows that had a poor body condition (score <2 on a five-point scale) were not included in the study.

On Day 109 of gestation, sows were moved to the farrowing room and were allocated into farrowing crates (1.95 m × 0.60 m) built with steel bars, which were centrally positioned in the farrowing pens (2.40 m × 1.80 m). Farrowing pens had a fully metal-slatted floor for sows and plastic-slatted floor for piglets. All farrowing pens had a metal heat pad at 36 °C and a supplemented heat lamp for the piglets during the first week of life.

The temperature in the farrowing room was kept constant, approximately at 22 °C, and light was on from 7:00 h to 18:00 h every day of the week. Sows were offered three meals per day (7:00 h, 13:00 h, and 17:00 h) and water was available ad libitum from drinking.

On Day 116 of gestation, at 7:00 h, farrowing was hormonally induced with 1 mL of Dalmazin^®^ in multiparous sows (D Cloprostenol 0.075 mg/mL, FatroIberica; Barcelona, Spain).

Treatments and manual interventions during farrowing followed the usual routine of the farm and were performed by the same person. When needed, intramuscular (IM) treatments were injected in the neck. When the interval time between two piglets exceeded 1 h and the cervical canal was dilated, 1 mL of Cabetocin (Decomoton^®^, Calier S.A.; Les Franqueses del Vallés, Barcelona, Spain) was injected. When the cervical canal was not sufficiently dilated, sows were treated with 200 mg of Vetrabutine Hydrochloride (Monzal^®^, Boehringer Ingelheim España, S.A.; Barcelona, Spain). When sows were very nervous to farrow, 5 mL of Azaperone (Stressnil^®^, Janssen Animal Health, Elanco; Brussels, Belgium) were administered.

Litter size was standardized by cross-fostering within 8 h post-farrowing, and piglets were weaned at twenty-one days old, according to veterinary recommendations.

### 2.2. Experimental Procedure 

It was necessary to see the sow’s face throughout the parturition without interference of the feeder. A vertical metal barrier (50 cm × 30 cm × 2 cm) was installed in the cranial part of the crate (in front of the sow’s feeder). Thus, the sow could not put her head under the feeder, and a complete vision of the sow’s face was guaranteed during parturition. One action camera (SK8 HD 4K; SK8 Urban, Spain) per sow was installed at 120 cm above the floor when the sow was lying down. The camera was focused on the sow’s face and its angle allowed us to record the rest of the body and the movements of the sow. When needed, an LED light was added to produce clearer and better-quality images. Video recordings were obtained from the beginning of the farrowing (before the first piglet’s expulsion) until the last piglet was born. If the sow was uncomfortable with the vertical metal barrier installed in front of the feeder, it was excluded from the study.

On Day 19 post-farrowing, sows were recorded again for 2 h between 8:30 h and 12:30 h, avoiding meals and farmer management, with the aim of obtaining control images. It was assumed that after more than two weeks of farrowing, there was no pain associated with farrowing, or its intensity was minimal.

For each sow, the following parameters were registered during farrowing by direct observations: the duration of farrowing, defined as the period of time between the first and the last piglet born; the time of expulsion of the piglet; the condition of each piglet at birth (born alive, stillborn, or mummified fetuses); and the number of treatments and manual interventions during farrowing.

### 2.3. Collection, Selection, and Processing of Images

One observer visualized all of the videos (partum and post-partum) and selected images in three different moments, always when the sow was lying laterally, and its face was fully visible:Pre-weaning (indicative of painless; Score 0). Images were chosen every 15–20 min.Interval time between two piglets’ expulsion (indicative of moderate pain; Score 1). One image was chosen in each interval time.Expulsion of the piglets (indicative of severe pain; Score 2). One image was chosen from within the 30 s prior to each piglet expulsion.

A total of 268 images were obtained (78 pre-weaning images; 91 images between two piglets’ expulsion; and 99 images before the expulsion of the piglet). Once all of the images were collected, they were cropped so only the faces of the sows were visible, to guarantee blinding by not revealing the rest of the sows’ body.

### 2.4. Development of the Facial Expression Scale

#### 2.4.1. Obtention of the Facial Expression Scale

A total of 268 images was assessed by the same observer (hereafter, the Silver Standard; SS). Images were randomly mixed and analyzed independently and blindly by the SS. For each image, SS determined five potential FAUs based on previous grimace score studies [4,5,8,11], which showed clear differences among the three moments studied. The five potential FAUs noticed in sows were: Tension above eyes, Snout angle, Neck tension, Temporal tension and ear position, and Cheek tension. Descriptions of the three moments of pain (score 0 = painless; score 1 = moderate pain; and score 2 = severe pain) of each FAU are explained in the Results section (Figure 1). The SS scored each FAU of all of the images using four categories: 0 (painless), 1 (moderate pain), 2 (severe pain), and IDK (I do not know or I do not feel confident assigning a degree-of-pain score).

#### 2.4.2. Inter- and Intra-Observer Reliability

A group of eight observers working as scientific researchers in the field of animal behavior and welfare at the Universitat Autònoma de Barcelona, without experience in studying animal facial expressions, were selected. Four observers (one man and three women) usually work in pig behavior, while the other four observers (one man and three women) work in other species such as cattle, companion animals, and zoo animals. The age range of the observers was between 28 and 37 years old. They were trained for 45 min by the SS concerning the physical differences among the three painful moments of the five FAUs in sows. The SS and observers evaluated 15 images together to standardize their assessment criteria. After that training, observers had to evaluate the five FAUs of 60 selected images for 45 s per image. Sixty images were randomly chosen from the 268 selected images: 18 images from the pre-weaning moment (score 0), 20 images from between two piglets’ expulsion (score 1), and 22 images from the moment of the piglet expulsion (score 2). Twelve of the 60 images were repeated twice to study intra-observer reliability. Observers were blind to the moment when images were taken, and they evaluated each FAU with the same four-point system used by the SS: 0 (painless), 1 (moderate pain), 2 (severe pain) or IDK (I do not know or I do not feel confident assigning a degree-of-pain score).

### 2.5. Statistical Analysis

Data were analyzed using the Statistical Analysis System (SAS 9.4 software, SAS institute Inc.; Cary, NC, USA). The significance level was established at *p* < 0.05. Descriptive values are given as mean ± SE.

Cohen’s Kappa coefficient was used to test the reliability between the SS assessment of the five FAUs and the moment when each image was obtained (pre-weaning, interval time between two piglets´ expulsion, or expulsion of the piglets). In addition, Spearman Correlation (r) was performed among the five FAUs (taking into account the SS´s assessment of 268 images).

Cohen’s Kappa coefficient (κ) was used to test intra- and inter-observer reliability. Inter-observer reliability was carried out by comparing the SS with the eight trained observers. Intra-observer reliability was accomplished by comparing repeated images per observer. The reliability among the following scores was analyzed: score 0 (painless), score 1 (moderate pain), score 2 (severe pain), and IDK (I do not know, introduced as a missing value). The level of reliability was categorized as follows: poor reliability (κ < 0.20), fair reliability (0.20 ≥ κ < 0.40), moderate reliability (0.40 ≥ κ < 0.60), good reliability (0.60 ≥ κ < 0.80), and very good reliability (0.80 ≥ κ < 1.00) [24].

In order to explore the effects of observer gender and their experience in pig behavior on the pain scores given in the evaluation session, two statistical analyses were performed. Firstly, for each FAU assessed in the evaluation session (scored as 0, 1 or 2), data were expressed as the number of FAUs with a score of 1 or 2 out of the total number of images assessed. Data were analyzed using generalized linear mixed models (GLIMMIX procedure) followed by a binary distribution. The model included the fixed effect of observer gender (women vs. men), observer experience in pig behavior (yes vs. no), and their interaction effect. The image was the experimental unit. The LSMEANS adjusted to Tukey’s honestly significant difference was used as a test of comparisons. Secondly, differences between inter- and intra-observer reliability means from the eight trained observers were evaluated using the univariate ANOVA test (PROC ANOVA). The fixed effects studied were observer gender (women vs. men), observer experience in pig behavior (yes vs. no), and the FAU assessed (from 1 to 5) and their interactions effects.

## 3. Results

### 3.1. Obtention of the Facial Expression Scale

Figure 1 shows the sow grimace scale with the five potential FAUs indicative of pain (Tension above eyes, Snout angle, Neck tension, Temporal tension and ear position, and Cheek tension) and the description of the three moments of pain (score 0 = painless; score 1 = moderate pain; and score 2 = severe pain) of each FAU. To determine which moment of pain the sow is in, at least one of the descriptions of each area must be observed clearly.

High reliability was obtained between the SS assessment of the five FAUs and the moment when each image was obtained (pre-weaning, interval time between two piglets’ expulsion or piglet expulsion). Very good reliability was obtained for Tension above eyes and Snout angle FAUs, and good reliability was obtained for Neck tension, Temporal tension and ear position, and Cheek tension FAUs (Table 1).

Studying the Spearman Correlations among the five FAUs (taking into account the SS assessment of 268 images), the highest positive correlations were found between Tension above eyes and Snout angle FAUs (r = 0.91), Tension above eyes and Neck tension FAUs (r = 0.87), Snout angle and Neck tension FAUs (r = 0.85), and Tension above eyes and Cheek tension FAUs (r = 0.84). All of the other FAUs showed positive correlations among each other (from r = 0.60 to r = 0.79) with *p* < 0.0001 in all pair correlations.

### 3.2. Inter- and Intra-Observer Reliability

Inter-observer reliability is summarized in Table 2. Good reliability was obtained for the Tension above eyes FAU. Moderate reliability was observed for the other FAUs studied.

Table 3 shows the intra-observer reliability of each FAU, analyzing 12 sow face images repeated twice during the 60-image evaluation. Neck tension was the FAU where the observers coincided in more cases with their own results, showing the best reliability. Tension above eyes and Snout angle FAUs also had good reliability. The Cheek tension FAU obtained a moderate reliability score and Temporal tension and ear position FAUs obtained a fair one.

### 3.3. Effect of the Observer Gender and Experience in Pig Behavior in Assessing Pain at Farrowing

Pain scores given by trained observers during the 60-image evaluation differed by gender.

Women scored a higher percentage of images as severe and moderate pain than did men in the Cheek tension FAU (severe pain: 51.9% vs. 45.5%, *p* = 0.0287 and moderate pain: 70.3% vs. 59.5%, *p* = 0.0302). Similarly, women scored a higher percentage of images as severe pain than did men in the Snout Angle FAU (47.9% vs. 27.7%, respectively*, p* = 0.0001) and Neck tension FAU (44.39% vs. 37.8%, respectively, *p* = 0.0392). However, pain scores given to the Tension above eyes and Temporal tension and ear position FAUs were not different by gender.

Pain scores given by trained observers were not affected by their experience in pig behavior or by the interaction effect of experience in pig behavior by gender.

The inter-observer reliability mean was affected by observer gender, as women observers obtained higher reliability with the SS (k = 0.47 ± 0.025) than did men (k = 0.38 ± 0.037) (*p* = 0.0318).

The inter-observer reliability mean was also affected by the FAUs, Tension above eyes (k = 0.63 ± 0.034) being the FAU with the best reliability with the SS, followed by Neck tension (k = 0.45 ± 0.026), Cheek tension (k = 0.42 ± 0.046), Snout Angle (k = 0.36 ± 0.038), and Temporal tension and ear position (k = 0.36 ± 0.031) (*p* = 0.0001).

Observer experience in pig behavior had no influence in the inter-observer reliability mean (k = 0.45 ± 0.039 from observers working with pigs vs. k = 0.44 ± 0.029 from observers working with other species, *p* = 0.8309).

The intra-observer reliability mean was also affected by observer gender. Women observers obtained higher reliability within images assessed twice (k = 0.62 ± 0.0524) than did men (k = 0.39 ± 0.0923) (*p* = 0.0254).

The intra-observer reliability was not affected by observer experience in pig behavior (k = 0.57 ± 0.0764 from observers working with pigs vs. k = 0.55 ± 0.0593 from observers working with other species, *p* = 0.8821).

## 4. Discussion

The present study´s results suggest that the proposed facial expression scale is a useful tool to assess pain in sows around farrowing. It has been demonstrated that a sow’s facial expressions change according to pain intensity and these expressions have been classified into three degrees of pain intensity: absence of pain, moderate pain, and severe pain. In fact, a substantial or near perfect reliability indicates that more than 80% of the images were scored as severe pain when they were obtained during piglet expulsion, and as moderate pain during the interval time between two piglets’ expulsion. These results are in agreement with Ison et al. [20], who were able to evaluate the intensity of pain around farrowing through the presence and frequency of corporal pain indicators in sows. They found that most of the corporal pain indicators were observed during farrowing, and considerably increased during piglet expulsion.

Five FAUs were identified by the SS, four of them have already been described in other species [4,5,6,7,8,9,10,11,12,13,14]. It should be noted that our study shares four out of five FAUs with the piglet grimace scale [8] (Tension above eyes, Snout angle, Cheek tension, and Temporal tension and ears), but has different FAUs compared to facial expression scales performed in other species [4,5,6,7,9,10,11,12,13,14]. The present work identified Neck Tension as a new facial expression zone never before described in animals. In previous studies of other species, almost all pictures were taken from the animals’ front, allowing the investigators to evaluate their whole faces. In our case, it was not possible to take an image from this angle, because the sows were lying laterally throughout almost all of the parturition. Therefore, we were able to evaluate the neck position. In addition, it could be that, for some investigators, the neck is not considered as a facial indicator. The neck is extended from the head, and its position can influence other FAUs expression, so it seems to be a good indicator that gives us valuable information. In fact, Häger et al. [14] described the head position in sheep as a pain indicator jointly with ear position. Head position is closely, though not completely, related to the neck position. There is a clear difference between the three neck pain descriptions, obtaining good reliability with the intensity of pain suffered by the sow. Based on the results of the present work, Neck Tension could be considered for study in other animal species as a potential facial expression zone indicative of pain.

The SS found that there is good or very good reliability in all FAUs studied, correlating the facial images with the painful moment that the sows are feeling. As shown in other studies [4,5,8,9,10,12], the Tension above eyes FAU (similar to the “Orbital Tightening” in other studies) showed the highest reliability in the sow’s facial expression scale. Tension above eyes not only evaluates the eye opening (orbital tightening), but also the eyebrow and its expression. In fact, we found a completely different pattern between Tension above eyes in sows at farrowing and the Orbital Tightening of the rest of the studies. The other animals´ facial scales [4,5,6,7,8,9,10,11,12,13,14] describe that the more pain the animal is suffering, the more tightly it closes its eyes; however, sows open them completely, partially showing the sclera. Di Giminiani´s [8] piglet study is the only other study that also described the eyebrow changes, agreeing with us that it becomes more curved when stronger pain was experienced by the animal.

As in other facial expression studies [8,9], inter- and intra-observer reliability analyses demonstrated good reliability with the SS and among themselves in the FAUs evaluation. Cheek tension and Tension above eyes were the best FAUs to assess pain in sows, as they were highly reproducible and highly predictive of pain status. Tension above eyes was a very good pain indicator in the rest of the facial expression studies [4,5,6,7,8,9,10,11,12,13,14]. Cheek tension was also a good FAU in other species such as sheep [9], mouse [4], rabbit [10], and ferret [13], where it is easy to observe cheek bulges when they are in pain. However, rat [5] and piglet [8] grimace scales also consider Cheek tension as a good pain indicator, but its interpretation is totally different. In both species, as with sows, bulging [5,8] occurs naturally when they eat or when they are relaxed, and this characteristic actually diminishes when the animals are in pain, flattening the cheek appearance. Temporal tension and ear position was the most imprecise FAU in sows, probably due to the ear size of sows and/or their position when sows were lying laterally (because they could cover parts of the face). In contrast, Ear position was a very good indicator in other species’ facial scales, being considered to be one of the best FAUs to evaluate pain in sheep [9,11], cats [25], rats [5], and mice [4]. There is even a horse grimace scale study [12] where all of the observers were able to evaluate horses’ ears.

There was a clear observer gender effect. As was seen in another study based on corporal pain indicators [16], women assessed significantly higher pain scores than did men when the sow was suffering, especially evaluating the moments with more intense pain, related to the piglet expulsion event. It could be because women commonly show a greater empathy towards human and non-human animals [16,26,27].

A lack of correlation was found between the observers’ swine production knowledge and the scores assigned to each FAU. Similar results were found in a previous study of facial piglet expression [8]. This expands the scale´s validity and its field of application. A short teaching period for all observers was enough to assure unified grading of facial expression changes and on the pain intensity suffered by the sow. Indeed, additional training is expected to ensure very good pain assessment.

Further studies are needed to validate this facial expression scale of pain during farrowing in sows. It could be carried out by correlating the sows’ facial expressions with the corporal pain indicators described by Ison et al. [20], which showed higher prevalence when the sow experienced stronger pain. Another way to validate the scale could be by non-steroidal anti-inflammatory drugs (NSAID) administration, comparing eutocic and dystocic farrowing, or comparing physiological indicators indicative of pain.

## 5. Conclusions

In conclusion, this is the first facial scale for pain assessment in sows during farrowing. This scale is accurate and easy to learn and could be easily applied in the field by veterinarians and farmers to detect animal suffering and improve animal welfare. Additionally, the reproducibility of the technique opens a broad field of pain research associated with several processes in swine.

## Figures and Tables

**Figure 1 animals-10-02113-f001:**
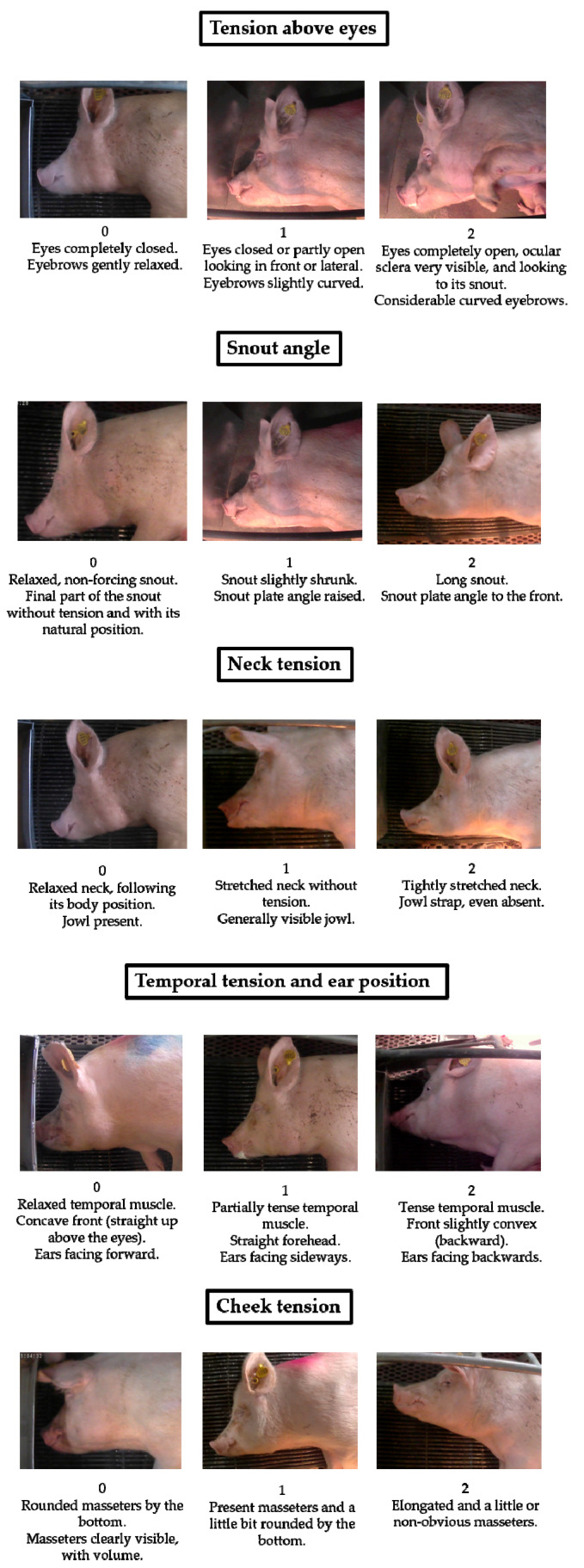
Sow grimace scale, with descriptions for each of the five facial action units (FAUs) employed: Tension above eyes, Snout angle, Neck tension, Temporal tension and ear position, and Cheek tension. FAUs are scored based on a three-grade scale: (score 0 = painless; score 1 = moderate pain and score 2 = severe pain).

**Table 1 animals-10-02113-t001:** Percentage of reliability and Kappa coefficient between images and the silver standard evaluation in each FAU of 268 sow face images: 78 images from before weaning (score 0), 91 images from the interval time between two piglets’ expulsion (score 1), and 99 images from piglet expulsion (score 2). The silver standard evaluation was defined as follows: painless moment (score 0), moderate painful moment (score 1), severe painful moment (score 2), and I do not know (IDK).

Image Moment	Silver Standard Evaluation	Tension Above Eyes	Snout Angle	Neck Tension	Temporal Tension and Ear Position	Cheek Tension
Before Weaning (score 0)	0	89.74	88.46	73.08	64.10	69.23
	1	10.26	11.54	19.23	34.62	16.67
	2	0.00	0.00	0.00	0.00	0.00
	IDK	0.00	0.00	7.69	1.28	14.10
The Interval Time between Two Piglets’ Expulsion (score 1)	0	0.00	1.10	2.20	6.59	3.29
	1	93.41	89.01	74.73	86.82	70.33
	2	6.59	9.89	13.19	6.59	8.80
	IDK	0.00	0.00	9.89	16.67	17.58
Piglet Expulsion (score 2)	0	1.01	0.00	1.01	3.03	1.01
	1	2.02	7.07	13.13	24.24	19.19
	2	96.97	92.93	78.79	72.73	67.68
	IDK	0.00	0.00	7.07	0.00	12.12
Cohen’s Kappa Coefficient		0.90 ***	0.85 ***	0.74 ***	0.63 ***	0.71 ***

*** *p*-value < 0.0001.

**Table 2 animals-10-02113-t002:** Inter-observer reliability (Kappa Coefficient) between the individual and mean evaluation of eight observers (from 01 to 08) and the Silver Standard of the five facial action units (FAUs) of 60 sow face images.

Observers									
FAU	01	02	03	04	05	06	07	08	Mean
Tension above Eyes	0.55	0.65	0.70	0.60	0.56	0.73	0.48	0.75	0.63
Snout Angle	0.24	0.55	0.33	0.50	0.33	0.37	0.27	0.31	0.36
Neck Tension	0.41	0.36	0.43	0.37	0.48	0.46	0.57	0.53	0.45
Temporal Tension and Ear Position	0.30	0.50	0.33	0.23	0.29	0.37	0.43	0.41	0.36
Cheek Tension	0.25	0.41	0.62	0.58	0.39	0.48	0.32	0.32	0.42

*p* < 0.0005 in all cases.

**Table 3 animals-10-02113-t003:** Individual and mean intra-observer reliability (Kappa Coefficient) of eight observers (from 01 to 08) for each facial action unit (FAU) of 12 sow face images repeated twice each during the 60-images evaluation.

Observers									
Pain Indicators	01	02	03	04	05	06	07	08	Mean
**Tension above Eyes**	0.09	0.71 ***	0.67 **	0.68 **	0.40 ***	0.89 ***	0.87 ***	0.65 ***	0.62
**Snout Angle**	0.73 ***	0.96 ***	0.81 ***	0.24 **	0.73 ***	0.77 ***	0.31 **	0.27 *	0.60
**Neck Tension**	0.22 *	0.87 ***	1 ***	0.40 *	0.87 ***	0.87 ***	0.75 ***	0.91 ***	0.74
**Temporal Tension and Ear Position**	0.07	0.91 ***	0.09	0.56 **	0.18	0.06	0.38 **	0.34 **	0.32
**Cheek Tension**	0.19 *	0.63 **	1 ***	0.26 **	0.40 ***	0.70 **	0.76 ***	0.21 *	0.52

*** *p* < 0.0005, ** *p* < 0.005, * *p* < 0.05.

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
