# Peer review of "Development of a Facial Expression Scale Using Farrowing as a Model of Pain in Sows"

_animals, 2020, doi:10.3390/ani10112113_

Round 1

Reviewer 1 Report

Pain assessment has always been an important element in assessing the feasibility of procedures in anesthetized animals. An interesting issue is the assessment of pain during various types of physiological activities, which include parturition and the course of labour. The conducted study seems interesting and constitutes to an important element for the development of various algorithms for pain assessment.

Attempts to assess pain based on image analysis are well known and links this work to the general trend of using non-invasive methods of assessing physiological and pathological phenomena.

An interesting addition would be the use of infrared thermography (IRT) which is a non-invasive imaging technique that allows to detect the radiant energy emitted by any object with a temperature above absolute zero. The radiated power detected by the thermal camera in the infrared spectrum is proportional to the fourth power of the object's absolute temperature and is used to calculate the temperature of the target e.g. the surface of the animal's body. Infrared radiation is often presented as a thermogram which is an image where the colour gradient corresponds with the distribution of

surface temperatures. If there are no such results, it is worth trying to supplement the

discussion with objective elements of pain perception based on the current literature. Pig research into the pain reaction during childbirth or other surgical procedures when the animal is under anaesthesia is carried out in the same scope

In my opinion, there is no objective pain assessment in the work. There are no biochemical, endocrine, or molecular tests that could objectify pain assessments.

If they were performed, the analysis should be supplemented with tests that objectively confirmed the actual symptoms of pain.

In summary, the work is interesting and contributes to further research using non-invasive methods. Visual assessment is useful but translated into the film image is distorted, therefore it is worth the work to have an element that does not raise any doubts to the extent and nature of the pain.

Author Response

-First of all, thank you for your revision and the nice words to our study. We would like to respond to all your comments and suggestions:

Pain assessment has always been an important element in assessing the feasibility of procedures in anesthetized animals. An interesting issue is the assessment of pain during various types of physiological activities, which include parturition and the course of labour. The conducted study seems interesting and constitutes to an important element for the development of various algorithms for pain assessment.

Attempts to assess pain based on image analysis are well known and links this work to the general trend of using non-invasive methods of assessing physiological and pathological phenomena.

An interesting addition would be the use of infrared thermography (IRT) which is a non-invasive imaging technique that allows to detect the radiant energy emitted by any object with a temperature above absolute zero. The radiated power detected by the thermal camera in the infrared spectrum is proportional to the fourth power of the object's absolute temperature and is used to calculate the temperature of the target e.g. the surface of the animal's body. Infrared radiation is often presented as a thermogram which is an image where the colour gradient corresponds with the distribution of surface temperatures.

Response 1: We really appreciate your suggestion about using infrared thermography, it could be very interesting and a useful tool to use in future studies.

If there are no such results, it is worth trying to supplement the discussion with objective elements of pain perception based on the current literature.

Response 2: To our knowledge, pain perception at farrowing have been assessed by Jarvis et al (1997) in sows. They suggested that nociceptive threshold increases during late pregnancy and parturition in the sow perhaps as an endogenous defence against labour pain. We didn’t introduce that reference in the manuscript as it doesn’t fit the objective of our study and may lead to confusion.

In order to give more “objective” indicators of pain at farrowing, we are introducing some examples of physiological indicators (see line 62-64).

Pig research into the pain reaction during childbirth or other surgical procedures when the animal is under anaesthesia is carried out in the same scope

Response 3: We agree, facial expressions research was previously developed on piglets. Despite of this, there are some morphological differences between piglets and sows. In fact, we found several differences between both productive ages (when we compared our study to those performed in piglets), and they are exposed on the discussion.

In my opinion, there is no objective pain assessment in the work. There are no biochemical, endocrine, or molecular tests that could objectify pain assessments.

Response 4: Ideally, pain in animals should be assessed by the combination of different indicators (productive, physiological and behavioural indicators). As first step, we developed the present facial scale. As second step, we propose different ways to validate it, that include the study of physiological indicators (see line 349).

Physiological indicators of pain are not enough specific, mainly for two reasons: (1) they only assessed the sensory component of pain (for instance, inflammation) and (2) other factors (such as stress or hormonal changes during parturition) could modified it. Behavioural indicators are considered more valid and specific for the study of pain in farm animals, as they could represent the both components of pain (sensory and emotional). Take into account that the emotional component of pain is important in order to obtain a real representation of the pain perception perceived by an animal. In addition, if behavior (or facial expressions) are assessed in a scientific way (blind observed, count of frequencies, study of repeatability…) is also considered an objective indicator.  For additional information we recommend you to see Weary et al., 2006; Viñuela-Fernandez et al., 2007, Sneddon et al., 2014 or the book Pain management in animals (chapter 4).

If they were performed, the analysis should be supplemented with tests that objectively confirmed the actual symptoms of pain.

Response 5: We agree with your observation about supplementing our study with other indicators of pain. Please, see line 345-349 in order to see which are the next steps proposed to do.

In summary, the work is interesting and contributes to further research using non-invasive methods. Visual assessment is useful but translated into the film image is distorted, therefore it is worth the work to have an element that does not raise any doubts to the extent and nature of the pain.

Reviewer 2 Report

In this manuscript by Navarro and colleagues, a grimace scale for sows is developed using farrowing as the pain stimulus. Although the scope of this finding is not huge (grimace scales for piglets having already been developed by two different groups), it is useful, and the finding of gender differences in scoring and accuracy is interesting too. The discussion is scholarly, although quite long for such a simple finding. My major complaint is that this would have been much stronger if the grimace scores were being compared to some other pain scoring system (like, e.g., the one used by Ison et al., 2016. Instead, the authors simply assume that pain is high during piglet expulsion, moderate in-between expulsion, and low/zero 19 days after farrowing. This is a reasonable way to develop a facial expression scale, but then it is common to compare it to something else, including other scales or other noxious stimuli. The authors may have been ethically prevented from attempting the latter, but they certainly could have done the former.

Other suggestions for improvement include:

  1. The sentence “Grimace scales are based on the…FACS, and were initially developed by Ekman and Friesen” implies that Ekman and Friesen developed grimace scales. Changing the “and” after the comma to “which” will solve the grammatical problem.
  2. Again in this study the claim that prey species tend to not express pain is made, citing McLennan. McLennan definitely said that, but to my knowledge the idea that prey species hide evidence of pain has never been actually shown to be true. It’s just asserted.
  3. Tables 2 and 3 could really use columns showing the average or median of the 8 scorers.
  4. The phrase “as far as the authors are concerned” has no place in a scientific paper.

Author Response

First of all, thank you for your revision and the nice words to our study. We would like to respond to all your comments and suggestions:

In this manuscript by Navarro and colleagues, a grimace scale for sows is developed using farrowing as the pain stimulus. Although the scope of this finding is not huge (grimace scales for piglets having already been developed by two different groups), it is useful, and the finding of gender differences in scoring and accuracy is interesting too.

Response 1: We agree, facial expressions research was previously developed on piglets. Despite of this, there are some morphological differences between piglets and sows. In fact, we found several differences between both productive ages (when we compared our study to those performed in piglets), and they are exposed on the discussion.

The discussion is scholarly, although quite long for such a simple finding.

Response 2: Discussion has been lightly reduced (see from line 292 to line 349).

My major complaint is that this would have been much stronger if the grimace scores were being compared to some other pain scoring system (like, e.g., the one used by Ison et al., 2016. Instead, the authors simply assume that pain is high during piglet expulsion, moderate in-between expulsion, and low/zero 19 days after farrowing. This is a reasonable way to develop a facial expression scale, but then it is common to compare it to something else, including other scales or other noxious stimuli. The authors may have been ethically prevented from attempting the latter, but they certainly could have done the former.

Response 3: We agree with your observation about supplementing our study with other indicators of pain. Our idea and the next step for this study is to analyze, continuously, the farrowing videos of these sows, notifying the behavioural pain indicators described by Ison et al., 2016 during farrowing and post farrowing, and correlate our facial expression scale with them.

Another idea that we thought is to do another study applying AINES, in order to see its effects on the intensity of the facial expressions.

Other suggestions for improvement include:

1. The sentence “Grimace scales are based on the…FACS, and were initially developed by Ekman and Friesen” implies that Ekman and Friesen developed grimace scales. Changing the “and” after the comma to “which” will solve the grammatical problem.

Response 4: Line 47. Thank you, we change the word “and” for “which” in order to solve the grammatical problem.

2. Again in this study the claim that prey species tend to not express pain is made, citing McLennan. McLennan definitely said that, but to my knowledge the idea that prey species hide evidence of pain has never been actually shown to be true. It’s just asserted.

Response 5: Line 59. We add the sentence “although it has not been proven, it seems….” So it doesn’t seem like such forceful affirmation.

3. Tables 2 and 3 could really use columns showing the average or median of the 8 scorers.

Response 6: Lines 238-251. Tables modified, we agree, they seem much easier to read and understand.

4. The phrase “as far as the authors are concerned” has no place in a scientific paper.

Response 7: Line 67. Phrase changed

-Line 351. Phrase removed